# Co-Optimization of Mechanical Properties and Radiopacity Through Radiopaque Filler Incorporation for Medical Tubing Applications

**DOI:** 10.3390/polym16223220

**Published:** 2024-11-20

**Authors:** Alan Nugent, Joseph Molloy, Maurice Kelly, Declan Mary Colbert

**Affiliations:** 1Innovative Polymer Compounds (IPC), Kilbeggan, N91 WF86 Westmeath, Ireland; 2PRISM Research Institute, Technological University of the Shannon, Athlone, N37 HD68 Westmeath, Ireland

**Keywords:** medical device, tubing, PEBAX, X-ray, radiopacity

## Abstract

Medical tubing, particularly cardiovascular tubing, is a critical area of research where continuous improvements are necessary to advance medical devices and improve patient care. While polymers are fundamental for these applications, on their own they present several limitations such as insufficient X-ray contrasting capabilities. As such, polymer composites utilizing radiopaque fillers are a necessity for this application. For medical tubing in vivo, radiopacity is a crucial parameter that virgin polymers alone fall short in achieving due to limited X-ray absorption. To address this shortcoming, inorganic radiopaque fillers such as barium sulphate (BaSO_4_) and bismuth oxychloride (BiOCl) are incorporated into polymer matrices to increase the X-ray contrast of the manufactured tubing. It is also known, however, that the incorporation of these fillers can affect the mechanical, physical, and thermal properties of the finished product. This research evaluated the impact of incorporating the two aforementioned fillers into Pebax^®^ 6333 SA01 MED at three different loading levels (10, 20, and 30 wt.%) on the physical, thermal, and mechanical properties of the composite. Composites were prepared by twin screw extrusion and injection molding followed by characterization of the mechanical (tensile, impact, and flexural), thermal (DSC), rheological (MFI), and physical (density and ash content) properties. The performed analysis shows that BiOCl enhanced the aesthetic properties, increased stiffness, and maintained flexibility while having minimal impact on the tensile and impact properties. When comparing BiOCl to BaSO_4_-filled composites, it was clear that depending on the application of the polymer composite, BiOCl may provide more desirable properties. The study highlights the importance of optimizing filler concentration and processing conditions to achieve desired composite properties for specific medical applications.

## 1. Introduction

### 1.1. Background

Medical devices play a crucial role in modern healthcare, providing critical support in diagnosis, treatment, and patient care. Medical device is a broad term for a range of products that cater to a vast array of solutions for healthcare issues. A medical device is any instrument or apparatus intended by the developer or manufacturer to be used for diagnosis, prevention, or treatment of diseases or disorders [1]. These devices range from basic devices to complex devices. These devices could be further categorized into the following: non-invasive devices that do not enter the body such as bandages and stethoscopes, and invasive devices that enter the body such as catheters and needle tips. Other devices include active devices that require a source of energy to function such as pacemakers and infusion pumps, and diagnostic devices used to diagnose diseases or conditions such as imaging devices (MRI, X-Ray, etc.) or dialysis machines.

As the global population continues to grow and age, development and innovation of new medical devices is of utmost importance. Many global leaders in medical device development and manufacturing are pushing the boundaries of materials to achieve the end goal of developing new innovative products for the healthcare market. There are many considerations when designing a medical device, an important consideration being material selection. Choice of materials will impact virtually all aspects of the device’s use: the dimensions, lifespan, and capabilities of the device; the cost and complexity of validation, manufacture, and application; and, most important of all, the physiological response of the patient and, thus, the viability of the device as a medical therapy [2].

Catheter tubing is a particularly interesting area of medical device development. Invasive catheter types include in-dwelling and intermittent catheters [3,4]. These catheters are used for a wide range of applications such as urinary catheters, central venous catheters (CVC), and pulmonary artery catheters (PAC). When reflecting on the design and development of specific devices such as catheter tubing, a major consideration is the materials from which the device will be produced. Not only is the polymer selection important, but also the ability to integrate radiopacity into the device, providing the surgeons with the ability to view the device during deployment under contrast X-ray-based techniques such as fluoroscopy, angiography, computed tomography (CT), and dual energy X-ray absorptiometry (DXA) [5].

### 1.2. Polymers in Medical Tubing

When developing medical tubing such as catheters, it is important to select the correct materials to ensure that the catheter performs as required. An appropriate combination of physical and mechanical properties is essential if a product is to be successful in its application [6]. The various types of catheters require different characteristics such as stiffness/flexibility, kink resistance, softness, smooth surface, elasticity, coefficient of friction, etc. Developments in innovative polymers are ongoing, where manufacturers are developing polymer groups with various different properties.

Synthetic polymers have extensive applications as biomaterials in medical implants. They can either be permanent, where their intended duration spans years, or temporary, where they are naturally biodegraded in vivo or removed upon healing [5]. The most common polymers used in catheter applications are thermoplastic polyurethane (TPU) [7], silicone [8], polyether block amide (PEBA) [9], and polyvinyl chloride (PVC) [10]. What makes these materials special is that manufacturers have the capabilities to tweak the formulations during synthesis to obtain materials with different properties. When reviewing materials available on the global polymer market, there are a wide variety of TPUs and PEBAs for example with different shore hardness and properties such as tensile modulus.

PEBA is one of the most versatile polymeric materials in use in the medical device industries. Poly (ether amide) are a group of thermoplastic elastomers (TPEs) that can be processed by injection molding and profile or film extrusion [11,12]. Pebax^®^ or poly(ether-*b*-amide) is a category of thermoplastic elastomeric copolymers prepared through block arrangement of hard polyamide and flexible polyether sections forming by condensation polymerization [13]. What makes Pebax^®^ so suitable for catheter applications is that while it shows excellent processability, it also displays high thermal and chemical stability [14]. The chemical structure of PEBA can be seen in Figure 1, where the hard segments are typically composed of PA12 and the soft segments have been identified as a PTMO in Pebax^®^ 2533 [15]. It is unclear if the same soft segment flexible polyether is used for all Pebax^®^ grades.

The various PEBA grades from Arkema in the Pebax^®^ range would have similar constituents, with differing compositions to achieve properties to suit particular applications. Pebax^®^ 6333 SA01 MED for example would, in theory, have a greater number of hard block PA 12 segments than Pebax^®^ 2533 SA01 MED to achieve a stiffer, harder material. Table 1 shows some of the various grades of Pebax^®^ SA01 MED for comparison purposes. It can be seen that the manufacturer of Pebax^®^, Arkema S.A., have developed a variety of PEBA thermoplastics suitable for a wide range of applications where either a hard or soft material is required, but also where a manufacturer may wish to focus on other properties such as tensile modulus (Young’s modulus) or impact strength. This has been possible due to the development of these grades by tuning the constituents prior to the condensation polymerization process.

### 1.3. Requirements for Radiopaque Materials

X-ray-computed tomography (CT) is a well-established tissue imaging technique employed in a variety of research and clinical settings. Specifically, CT is a non-invasive clinical diagnostic tool that allows for 3D visual reconstruction and segmentation of tissues of interest [20,21,22]. Lusic and Grinstaff, (2013) summarize that, as a rule, materials possessing higher density (ρ), or a high atomic number (Z), tend to better absorb X-rays. The relationship is best expressed in Equation (1) [20] for X-ray absorption coefficient (μ):(1)$$\mu \approx \frac{\rho Z^{4}}{AE^{3}}$$
where

A= atomic mass;E= X-ray energy.

Consequently, X-ray attenuating contrast media containing atoms of high atomic number (most commonly iodine or barium) are frequently used in clinical settings to obtain images of soft tissues.

### 1.4. Inorganic Fillers

When a radiopaque material is used for the single-scan method, it must be homogenous and uniform in appearance and must not create scatter. One such material is barium sulphate, which is commonly used for gastrointestinal (and cardiovascular) imaging [23,24]. Ingestion of certain forms of barium (e.g., barium carbonate or barium fluoride) in toxic amounts can lead to gastrointestinal (vomiting, diarrhea, and abdominal pain), cardiac and skeletomuscular stimulation followed by paralysis [25,26]. The toxicity of barium compounds depends on their solubility. Barium sulphate (BaSO_4_), which is often used for medical purposes, remains essentially unabsorbed and is unlikely to cause adverse effects [25]. Chemically pure BaSO_4_ is nontoxic to humans, due to its insolubility, and is frequently used as a contrast agent in the X-ray diagnosis of colorectal and upper gastrointestinal examinations [27]. BaSO_4_ is commonly used in medical devices, incorporated homogeneously into a polymer suitable for a particular application. It is typically incorporated at levels of 10–40 wt.% but is known to be incorporated up to 60 wt.% [28].

Bismuth compounds are considered to be poorly to moderately absorbed after inhalation or ingestion, but there are no quantitative data. The highest concentration of absorbed bismuth is typically found in the kidney and liver, and generally excreted through the urine [29]. Similar to BaSO_4_, bismuth oxychloride is insoluble in water, reducing the toxicity of the compound. BiOCl is commonly used in medical devices, incorporated homogeneously into a polymer suitable for a particular application. Similar to barium sulphate, bismuth oxychloride is typically incorporated at levels of 10–40 wt.%. High-loading filler fractions up to 60 wt.% are often required to achieve good X-ray visibility, although such amounts of radiopaque filler can strongly influence mechanical properties [30].

Other radiopaque additives such as bismuth subcarbonate (Bi_2_OCO_3_), bismuth trioxide (Bi_2_O_3_), and tungsten (W) each yield their own advantages and disadvantages. While Bi_2_OCO_3_ is the most popular radiopaque filler after BaSO_4_ due to its greater radiopacity, its polymer compatibility is limited and is unstable at temperatures in excess of 200 °C. Bi_2_O_3_, similar to Bi_2_OCO_3_, has excellent radiopacity, however, its yellow appearance may be undesirable, especially when browning occurs at elevated temperatures. Tungsten is different from the other fillers; due to its greater density (19.28 g/cm^3^), it is generally loaded at levels in excess of 80 wt.% [31]. While the occupied volume is similar to those of barium sulphate and bismuth oxychloride, it is an extremely dense compound. It does, however, have excellent radiopaque properties. That being said, the appearance of the compound is dark for those who desire more colorful devices, and may not be an option. Some advantages and disadvantages of both BaSO_4_ and BiOCl can be seen in Table 2.

**Table 2 polymers-16-03220-t002:** Advantages and disadvantages of barium sulphate and bismuth oxychloride [32].

Filler	Advantages	Disadvantages
Barium sulphate (BaSO_4_)	Widely used in industry Relatively inexpensive Very process stable Easy to color	High loading levels Poor tinting strength
Bismuth oxychloride (BiOCl)	Excellent white color Highly compatible with wide range of polymers Smooth surface finish	Difficult to color Susceptible to UV degradation

## 2. Materials and Methods

### 2.1. Materials

Polyether block amide (Pebax^®^ 6333 SA01 MED), barium sulphate (Blanc Fixe XR HN), and bismuth oxychloride (Bismuth Oxychloride BPC) were obtained from National Chemicals Company (NCC) (Dublin, Ireland). The properties of the Pebax^®^ 6333 SA01 MED can be seen in Table 3. The BaSO_4_ used had a density of 4.5 g/cm^3^ and an average particle size of approximately 1 µm. The BiOCl used had a density of 7.7 g/cm^3^ and an average particle size of approximately 1.5 µm. Both fillers are white in appearance.

### 2.2. Preparation of PEBA/Filler Composites

Pebax^®^ 6333 SA01 MED was dried at 70 °C for 4 h prior to compounding on the twin-screw extruder, while the Blanc Fixe XR HN and bismuth oxychloride BPC were conditioned at 20 °C for 24 h prior to compounding. The moisture content of each of the materials was checked prior to compounding using a Brabender Aquatrac+ moisture analyzer (Brabender, Duisburg, Germany), where the Pebax^®^ 6333 SA01 MED and the fillers were dried to below 0.08 percent moisture (%H_2_O). The composites were then dried at 70 °C for 4 h until a moisture content of <0.08%H_2_O was achieved. The blend compositions manufactured via hot-melt extrusion are detailed in Table 4.

### 2.3. Twin-Screw Compounding

All melt processing for the production of PEBA/filler composites for the purposes of this study were carried out on a Leistritz ZSE 27MAXX co-rotating twin-screw extruder (Leistritz Extrusionstechnik GmbH, Nürnberg, Germany), composed of 27 mm diameter mixing screws and a 40 L/D ratio. The extrusion temperatures utilized are shown in Table 5. The rotational speed of the co-rotating screws was kept constant at 310 RPM. The torque required to obtain maximum dispersion of the filler increased from 70% torque for the 10 wt.% filled composites to 75% torque for the 30 wt.% filled composites. The extruder utilized Coperion K-Tron gravimetric feeders (Coperion GmbH, Stuttgart, Germany), allowing for highly accurate dosing of each element with a feed rate of 25 kg/h. The twin-screw extruder used for the trials is the property of Innovative Polymer Compounds (IPC) situated in Kilbeggan, Ireland. Upon material exiting the dies of the machine, strands were cooled in a water bath set at 10 °C before pelletizing using a Rieter Primo E strand pelletizer (MAAG Group, Großostheim, Germany) to produce pellets 2.5 mm in diameter and 3.0 mm in length.

### 2.4. Injection Molding

Injection molding of impact and tensile test specimen was carried out on an Arburg Allrounder 370E Golden Electric injection molding machine (ARBURG GmbH + Co KG, Lossburg, Germany) located in Ross Polymer Services Ltd., Athlone, Ireland. The Arburg has a maximum clamping force of 600 kN, a screw diameter of 30 mm and a theoretical stroke volume of up to 85 cm^3^. The machine has four thermocouples along the barrel and one on the nozzle controlling the input temperature in the process. The recommended processing temperatures from Arkema were used in the processing of the composites, 230 °C in zone 1 to 260 °C at the nozzle. A mold temperature of 40 °C was maintained by means of an a Wittmann Tempro basic (Wittmann Technology GmbH, Vienna, Austria) temperature controller. The mold temperature was confirmed using an ATP temperature probe. The mold utilized for in the production of tensile and impact test specimen was a ‘two by two’ mold, with cooling performed for 15 s (s). The geometry of the molded samples can be seen in Figure 2.

### 2.5. Mechanical Analysis

#### 2.5.1. Tensile Testing

Tensile testing was performed in accordance with EN ISO 527-1 international standard for test specimens using a Zwick Roell Z005 tensile tester (Zwick GmbH & Co. KG, Ulm, Germany). A total of 6 test specimens for each sample were tested with dimensions of 112 mm in length, 6 ± 0.2 mm in width, and 3 ± 0.2 mm in thickness. The tensile testing was performed at ambient room temperature, approximately 20 °C. The tensile properties from the trials were obtained directly from the machine software. No additional calculations were required.

#### 2.5.2. Flexural Testing

The flexural tests were performed using a three-point bending configuration by mounting a dedicated tool on the universal testing machine (Walter + Bai LFV 300, Walter + Bai AG, Löhningen, Switzerland). The radius of the support rollers and the loading nose was 25 mm and the distance between the centers of the rollers was 60 mm. The test samples were prismatic (bar-shaped samples), and each testing set comprised 6 specimens. The specimen can be seen in Figure 2. The test speed was kept constant at 5 mm/min. The mechanical tests were conducted at ambient room temperature, approximately 20 °C and 50 ± 5% relative humidity (rh).

#### 2.5.3. Impact Testing

A calibrated Zwick Roell CEAST 6245 (Zwick Roell, Elm, Germany) was used to carry out the Charpy notched impact test on prismatic-shaped samples, and each test set comprised 5 specimens with dimensions of 123 mm in length, 12.3 ± 0.2 mm in width, and 6 ± 0.2 mm in thickness. The specimens were notched to a depth of 1.0 mm and placed in the sample holder with the notched placed centrally with the notch facing away from the falling pendulum arm. The 4 Joule (J) pendulum hammer was then locked in the upward position and subsequently released. Upon impact of the downward swinging pendulum, the impact energy imparted into the specimen is recorded in J. The impact strength of the samples was calculated using Equations (2) and (3).
(2)K=m×g×(H−h)
where

*K* = notched impact energy;*m* = mass of the hammer;*g* = gravity constant;*H* = initial height of pendulum hammer;*h* = distance travelled by pendulum hammer post impact.(3)$$\alpha =\frac{K}{A}$$
where
*α* = Charpy notched impact (J/m^2^);*A* = cross-sectional area minus notch (m);*g* = gravity constant;*H* = initial height of pendulum hammer;*h* = distance travelled by pendulum hammer post impact.

### 2.6. Thermal Analysis

#### 2.6.1. Melt Flow Index

MFI analysis was conducted on CEAST 7025,000 (Instron CEAST, Torino, Italy). As the melt flow of the polymer is extremely sensitive to moisture, which can affect the results of the melt flow, each sample formulation was dried to the same moisture content, <0.08%H_2_O. The melt flow apparatus uses Instron melt flow modular software, which calculates the MFI value in g 10 min^−1^ automatically, reducing any risk of human error that can occur when incorporating the manual material cutting method. Once the software was prepared with parameters for each sample such as product name and density, the barrel was carefully filled with granules from the specific testing batch, ensuring to press the material the reduce any voids or air pockets within the barrel. The piston was then placed in the barrel, with the displacement measurement sensor in place. Once the test had concluded, the results were graphed with the MFI values for each sample.

#### 2.6.2. Differential Scanning Calorimetry

DSC analysis was carried out using a Pyrus 6 DSC (PerkinElmer, Waltham, MA, USA), which was calibrated using indium as the reference material. Samples between 6–8 mg were accurately measured and placed in lid-sealed aluminum pans. Calorimetry scans were performed using a heating/cooling rate of 10 °C min^−1^ applying standard heat from 20–250 °C min^−1^ for all samples. Samples were tested under nitrogen atmosphere with a flow of 30 mL min^−1^ to avoid oxidation. The resultant thermograms and determination of thermal transitions was performed using TA Universal Analysis software. The degree of crystallinity of the hard-crystalline phase of the PA in Pebax^®^ was calculated using Equation (4).
(4)$$X_{c}=\frac{{∆H}_{f}}{{∆H}_{f*}}\times 100$$
where
*∆H_f_* = enthalpy of fusion obtained from thermogram;*∆H_f*_* = enthalpy of fusion of 100% crystalline PA12 in the hard regions of the PEBAX [37].

### 2.7. Physical Analysis

#### 2.7.1. Density

Prior to performing density analysis, the theoretical density of composite samples was calculated using Equations (5) and (6).
(5)$$\left(\frac{M_{f1}}{\rho_{f1}}=V_{f1}\right)+\left(\frac{M_{f2}}{\rho_{f2}}=V_{f2}\right)+\ldots \left(\frac{M_{fn}}{\rho_{fn}}=V_{fn}\right)=V_{tot}$$
(6)$$\frac{M_{tot}}{V_{tot}}=\rho_{theor}$$
where 

*M_f_* = mass fraction;*ρ_f_* = density of mass fraction;*V_f_* = volume fraction;V_tot_ = total volume;M_tot_ = total mass of the composite;*ρ_theor_* = theoretical density.

Determination of the density of the various samples is a manual process, whereby the polymer granules from each formulation are heat-pressed into a square plaque with dimensions of 50 ± 0.2 mm in length, 50 ± 0.2 mm in width, and 2 ± 0.2 mm in thickness. A Kern ADJ 200-4 (KERN & SOHN GmbH, Balingen, Germany) digital balance accurate to 0.1 mg was used for weighing the samples for density. The pre-prepared samples are placed on the measurement plate of the scales, with the value recorded after stabilization in excel as ‘W1’. The ‘pan straddle’ apparatus was then placed into the measurement section of the scales, with the immersion vessel placed on a self-supporting tripod. The cradle was then attached to the pan straddle, and the immersion vessel filled with boiling water. Once the temperature of the water had reached 20 °C, the scales were zeroed, and the same polymer sample was placed on the cradle submerged in water. This was allowed to stabilize over 5 min to provide an accurate measurement. This value was then input into excel as ‘W2’. Density of samples were subsequently calculated using Equation (7).
(7)$$\rho =\frac{W_{1}}{W_{1}-W_{2}}$$
where 

*W*_1_ = mass of sample in air;*W*_2_ = mass of sample underwater.

#### 2.7.2. Ash Content Analysis

Ash content analysis is a measurement of filler content of a polymer composite, typically used as an alternative to thermogravimetric analysis (TGA). The main difference between ash content analysis and TGA is that ash content analysis does not continuously monitor and graph the results from the test. The test was carried out using a SNOL 3/1100 (Umega Group, Ukmergė, Lithuania) muffle furnace, whereby a porcelain crucible was placed into the muffle furnace for 15 min, removed, and placed into the desiccator to cool. The crucible was then weighed using the Kern ADJ 200-4 digital balance. The value obtained was recorded in excel as ‘W_1_’. Next, 2 g of the polymer sample was then weighed in the second crucible, the crucible and polymer were then weighed and recorded as ‘W_2_’ in excel. The crucible with the 2 g of polymer was placed into the muffle furnace. The muffle furnace was set to 850 °C, with the polymer sample heating for 10 min. After the 10 min had lapsed, the crucible was removed and placed in the desiccator to cool. After cooling, the weight of the crucible and polymer was recorded in excel as ‘W_3_’. These steps of heating the polymer and cooling were continued until a constant weight of the crucible and polymer was achieved, or until the variation was <0.002 g. The ash content was subsequently calculated using Equation (8).
(8)$$Ash,mass\%=\frac{W_{3}-W_{1}}{W_{2}-W_{1}}\times 100$$
where 

*W*_1_ = mass of crucible (g):*W*_2_ = mass of polymer sample and crucible together (g):*W*_3_ = mass of ash sample and crucible together (g).

#### 2.7.3. X-Ray Medical Imaging

X-ray medical imaging is an imaging technique similar to fluoroscopy; where fluoroscopy provides real-time dynamic illustration, X-ray provides static images. X-ray imaging was carried out on a Phillips Azurion 7 M20 digital fluoroscopy machine (Medical & Engineering Technologies ATU, Galway, Ireland). Films of 0.25 mm thickness, approximately 50 mm × 50 mm, were prepared for each formulation using a Polystat 200T (Servitec Maschinenservice GmbH, Berlin, Germany) hydraulic press. For the film preparation, the injection-molded samples for the respective formulations were placed between stainless steel plates and PTFE film. The hydraulic press was set to 200 °C with a hydraulic pressure of 38 bar.

The respective samples were labelled before X-ray, and placed in order on the imaging table. The X-ray mode was selected; for this study, the head scan mode was used to give a crisp, clear image. The radiation dose used for the imaging was 41 kV.

## 3. Results and Discussion

### 3.1. Processing Observations

#### 3.1.1. Twin-Screw Extrusion

Observations were noted throughout the extrusion trials. These included the screw speed, machine torque, melt pressure at the die, and polymer melt temperature. The data recorded from the compounding trials were graphically represented and can be seen in Figure 3. All graphs show the average value obtained throughout each filler level of both BaSO_4_ (orange) and BiOCl (green). While the BaSO_4_ remained most consistent throughout compounding, there were no issues in processing. During the compounding of both PEBA/BaSO_4_ and PEBA/BiOCl, it can be seen in Figure 3B that there was an increase in torque as the filler level increased. PEBA/BaSO_4_ showed a slight increase in torque 69% to 70% torque, while the PEBA/BiOCl display a greater increase, 73% to 77% to 75% torque. It can be seen in Figure 3A that the screw speed was reduced as the filler level increased for each respective filler. It was necessary to reduce the screw speed in an attempt to maintain a consistent torque level. This increase in torque may be due to the addition of polymer and dry filler in the first zone of the extruder. This may put additional pressure on the screws, leading to an increase in torque required by the motor to rotate screws. The melted blend may occur; however, it is expected that the viscosity of the composite may reduce due to the shear occurring further in the process. It can be seen in Figure 3C and D that the melt pressure and temperatures were both consistent. Figure 3D shows that the melt temperatures in the various PEBA/BaSO_4_ composites were lower melt temperature than those of PEBA/BiOCl. This may suggest that the BiOCl has a higher thermal conductivity value than BaSO_4_, where the PEBA/BiOCl composites absorb more heat from the extrusion process. At the highest filler loading level (30 wt.%), the melt temperatures of the two formulations were essentially superimposable. It was posited that this was due to the increased volume of BaSO_4_ in the composition compared to BiOCl. Though both materials were loaded at 30 wt.%, the density of the BaSO_4_ is 4.5 g/cm^3^, whereas the BiOCl has a density of 7.8 g/cm^3^; therefore, BaSO_4_ occupies 42% more volume in the melt.

Figure 4 shows the appearance of pellets pre- and post-extrusion, where (A) is the Pebax^®^ SA01 MED, (B), (C), and (D) are PEBA/BaSO_4_-10, -20, and -30 respectively, while (E), (F), and (G) are PEBA/BiOCl-10, -20, and -30 respectively. The ‘whiteness’ of the composites is obvious from the image. While it is clear that both PEBA/BaSO_4_ and PEBA/BiOCl obtain a deeper white color as the filler level increases, the PEBA/BiOCl has a more pronounced white color. It is evident that at the lower loading levels, the PEBA/BaSO_4_ composites are much more translucent than the lower loadings of PEBA/BiOCl. Both the PEBA/BaSO_4_-30 and PEBA/BiOCl-30 displayed a brilliant white color, however, the PEBA/BiOCl-30 displays a somewhat yellow hue.

#### 3.1.2. Injection Molding

Similar to the extrusion trials, there were a number of observations made during the injection-molding trials. Two parameters of interest during injection molding were melt cushion and plasticizing time. Figure 5 depicts the average values obtained from the injection-molding trials, with Figure 5A displaying the average melt cushion and Figure 5B the average plasticizing time. It can be seen that the trends are similar between the PEBA/BaSO_4_ and PEBA/BiOCl, where there is a clear increase in the melt cushion as the filler addition level increases for both PEBA/BaSO_4_ and PEBA/BiOCl. In relation to the plasticizing time for the various filler levels of PEBA/BaSO_4_ and PEBA/BiOCl composites, it was clear that as the loading increased, there was a reduction in plasticizing time. This may be as a result of the reduction in viscosity, which is a major variable in the plasticizing time in injection molding [38]. An observation was made whereby as the filler content increase, so too did the melt flow index. This may have been due to the shear in the extrusion process, but the viscosity was reduced. In relation to the melt cushion, as materials with higher densities have less compressibility, a smaller volume of melt cushion would be required to maintain consistent pressure in the mold to obtain dimensionally stable components. In contrast to this, with shear in the extrusion processes prior to injection molding, the molecular weight of the various composites is expected to have reduced, which leads to a lower viscosity, therefore requiring a larger melt cushion to provide more control during the packing phase. It was observed that initially, at the lower loading levels of inorganic filler for both PEBA/BaSO_4_ and PEBA/BiOCl composites, the melt cushion had dropped, but increased again as the filler loading within the composites increased. It could be said that due to the increase in the melt density as the filler content increases, the melt cushion increases to compensate for the loss in volume of the composite at higher filler loading. The standard deviation of each of the process parameters explains a lot about the behavior of the material. Consistency, or low standard deviation, shows that the material is consistent, with little variation throughout. The consistency of the melt cushion and plasticizing times for each of the composites was crucial in evaluating the dispersion of the filler throughout the respective polymer matrix. Without consistency in the process, this would signify a lack of consistency in the composites themselves.

### 3.2. Mechanical Properties

#### 3.2.1. Tensile Properties

There are multiple factors that may influence the tensile properties of a polymer or polymer composite. Bayazian and Schoeppner found that when processing a polymer using extrusion technology, the shear and other stresses that occur in the process causes chain scission to occur, reducing the molecular weight of the polymer [39]. This reduction in molecular weight can cause the mechanical properties of the polymer to be reduced. Another factor that plays a major role in the mechanical properties of polymer composites is a phenomenon known as nucleation. With the incorporation of micro (>1 µm) or nano (1–100 nm) particles into a polymer matrix, it is not uncommon to see an increase in the degree of crystallinity of the polymer due to a nucleating affect brought on by the filler. This can lead to an enhancement of the tensile properties of the polymer due to improved organization of lamellar crystals. Composite strength and toughness are very much dependent on the adhesion quality, the particle size, and the loading levels in the polymer [40]. Table 6 displays the tensile properties of the various PEBA/BaSO_4_ and PEBA/BiOCl composites in comparison to the Pebax^®^ 6333 SA01 MED (PEBA100).

From the obtained data, it is evident that BiOCl is a more effective reinforcement for tensile properties than BaSO_4_. The incorporation of BiOCl caused minor increases in the tensile strength at higher loading levels of 20 and 30 wt.%, showing increases of 2.24% and 2.21%. Conversely the BaSO_4_ led to a consistent decrease in the tensile strength of the composites immediately upon loading with the 12, 20, and 30 wt.% loadings resulting in a 7.11%, 7.54%, and 17.35% decrease, respectively. With respect to the percentage of elongation of the composite samples, all loadings of filler resulted in a reduced elongational ability while the tensile modulus displayed the inverse trend (i.e., increasing with respect to increased filler content). Such results are indicative that the composite materials are becoming increasingly stiffer and less ductile, a common observation with filler/polymer composites. The increase in tensile modulus observed in the BiOCl and BaSO_4_ composites is likely due to the inherent rigidity of the fillers themselves, which leads to enhanced rigidity of the overall system. The reduction in tensile strength and elongation may arise again due to the aforementioned point but so too due to a lack of interfacial adhesion between the filler particles and PEBA matrix. Additionally, there may be filler–filler interactions occurring at the higher wt.% of BaSO_4,_ leading to the formation of agglomerates within the composite [41]. Further interactions between filler and matrix may lead to hindered rearrangement of the polymeric chains, which has a negative impact on the recrystallisation of the material, thus leading to a decrease in tensile strength [42]. The particle shape of the filler may too cause an adverse impact on the tensile properties of composites. At low loading levels, the effect of particle shape is negligible but the impact becomes more pronounced at loading levels exceeding 20%. BiOCl has platelet-shaped irregular-edged particles [43], whereas BaSO_4_ has spherical particles [44]. Paknia et al. demonstrated using finite element analysis that circular-shaped particles result in a less stiff composite than shapes of more defined structure [45]. It may be posited, therefore, that the more structured particle shape of the BiOCl limits the movement of dislocations around the boundaries of the particles compared to the spherical BaSO_4_ particles, resulting in the enhanced tensile performance.

#### 3.2.2. Flexural Properties

The flexural behavior is similar to the tensile behavior of a material, but rather than the resistance to stretching as is the case with tensile, the flexural behavior of a material is a measure of its resistance to bending deformation. This measurement of stiffness provides in-depth knowledge about a material and how it behaves when a simple beam load is exerted upon the material and can be comparable to the stiffness observed in the Young’s modulus. Figure 6 and Table 7 display the various flexural properties of PEBA100 and the PEBA/BaSO_4_ and PEBA/BiOCl composites.

It can be seen from the graphs that the composites perform very well in comparison to the polymer, whereby in both cases of flexural modulus (Figure 6—blue) and flexural strength (Figure 6—red), there is a relatively linear increase in value with respect to filler loading level. It is evident from the obtained data that while both fillers improve upon the flexural properties compared to the virgin polymer, there is a negligible difference in improvement when comparing the two fillers. The flexural (or bending) modulus is a quantifying factor of how much a given sample will bend when a given load is applied to the sample [46], with a higher value of flexural modulus being indicative of a stiffer material. As is evident from Figure 6, as the filler content increases, so too does the stiffness of the sample. It can be seen that as the filler content increases in the composites, the materials become much stiffer. While the differences in flexural moduli are miniscule when comparing the composites to the polymer, it is seen that the PEBA/BiOCl composite is stiffer at the higher loading than the PEBA/BaSO_4_. It is clear that with the addition of inorganic filler to the polymer matrix, the composites become much stiffer. This is exemplified not only by the tensile strength and Young’s modulus, but also by the values of the flexural strength and modulus. As seen with the flexural strength, as the filler level increased, so too did the flexural strength. The effect of stiffness increasing with respect to filler loading level is well-documented in the literature. Taş and Soykok demonstrated an increase in flexural modulus of 23.7% when utilizing carbon nanotubes and carbon fiber loaded into PEEK at 0.5 wt.% [47]. Ramesh et al. demonstrated an increase in flexural modulus from 87.81 GPa for virgin epoxy resin to 156.65 GPa when incorporating basalt/jute/banana/carbon fabric as a filler [48]. Arulmuruan et al. utilized BaSO_4_ as a filler to improve the mechanical properties of natural hybrid-reinforced composites. The authors noted that the stiffness of the composites increased with respect to BaSO_4_ inclusion levels, due to the materials resistance to crack initiation and propagation [49]. Bociong, et al. made similar observations, whereby at 40 wt.% silanized silica-filled dental resin, the flexural strength increased. However, as the filler loading increased further, there was a noticeable reduction in flexural strength [50]. This can be associated with the higher volume of filler in the matrix creating a lack of interfacial adhesion between filler and polymer, causing weak areas around the filler.

#### 3.2.3. Impact Strength

It was important to analyze the effects of incorporating inorganic filler at the various loadings on the impact strength or rigidity of the composites. In order to evaluate this, each sample was subjected to Charpy notched impact testing. The impact values obtained for the PEBA100 and the various composites can be seen in Figure 7. It is clear from the graph that while there is a small increase in the impact strength of the filled PEBA grades (4.7–5.33% increase) in comparison to the PEBA100, and there is an observed reduction in impact strength between PEBA100 and PEBA/BaSO_4_30. While the incorporation of inorganic filler into the PEBA increases the impact strength, it must be noted that there is a point whereby the addition of filler causes the polymer to become too brittle, leading to catastrophic failure. This was the case for PEBA/BaSO_4_30 (33.88 kJ/m^2^), where the samples broke, requiring much less energy to achieve the break than the energy absorbed by the PEBA/BaSO_4_10, PEBA/BaSO_4_20, and all PEBA/BiOCl composites. The breakage of PEBA/BaSO_4_30 demonstrates the volumetric overloading of inorganic filler, leading to a brittle composite. There was minimal difference between the PEBA/BiOCl composites of various filler loadings, showing that there was no clear reduction in impact strength as the loading increased, contrary to the results obtained for the PEBA/BaSO_4_-30. A case could be made for the PEBA/BiOCl composites, where due to the higher density of the BiOCl compared to BaSO_4_, the volume of BiOCl in the PEBA/BiOCl-30 composite does not cross into the brittle point as the PEBA/BaSO_4_30 does.

An important factor in the toughness of a polymer composite is the not only the addition level of filler, but also the particle size and particle–matrix interaction. Sreekanth et al. conducted research on the effects of particle size and filler level on the mechanical properties of TPE composites. It was established that larger particle sizes led to a reduction in the toughness of the composites. Similarly, as the filler content increased, there was a clear reduction in toughness of the composites. It was summarized that the greater the addition of filler, the greater the reduction in elasticity and ability of the composite to absorb energy, reducing the deformability of the polymer matrix [51]. In a similar study, Lakkundi et al. confirmed the findings of Sreekanth et al., whereby an increased loading of filler in the polymer caused a weaker interfacial interaction between polymer and filler due to the high loading of filler, leading to a less ductile composite [52].

### 3.3. Thermal Analysis

#### 3.3.1. Melt Flow Analysis

The rheological properties of the polymer and polymer composites are crucial information for processing methods such as injection molding, and tube extrusion. Figure 8 displays the melt flow index of PEBA100 and the various PEBA/BaSO_4_ and PEBA/BiOCl composites. It is clear from the graph that, irrespective of filler content, the MFI has increased for all composites. This shows that the shear as a result of the twin-screw compounding has a more pronounced effect on the flowability of the composites in comparison to the filler content. The increase in melt flow of the PEBA/BaSO_4_ composites with respect to PEBA100 were 11.9%, 17.5%, and 21.8% with the respective increases in filler loading. Similarly, the PEBA/BiOCl composites experienced increases of 13.5%, 16%, and 25% with the respective increases in filler loading. There was minimal difference in melt flow between the PEBA/BaSO_4_ and PEBA/BiOCl composites, showing that the % volume of filler in the composites had minimal impact on the melt flow.

While it may be expected that the composites viscosity would increase with the incorporation of large volumes of filler, this is clearly not always the case. There are other factors that influence the rheological characteristics of the composite. While the addition of fillers into the polymer matrix may enhance certain properties of the polymer, the rheological properties are not as straightforward. The additional processing step, and, therefore, thermal history, adds additional shear into the polymer, reducing the molecular weight, which leads to a reduction in viscosity, or increase in melt flow. This shear, along with adequate mixing, can lead to a very well-dispersed filler within the composite, which leads to a reduction in viscosity. If the mixing was inadequate, agglomerates within the polymer matrix would lead to an increase in viscosity. A case could be made for excessive shear in the polymer composites, which may have caused an excessive breakdown of molecular weight of the composites. Hnatkova et al. analyzed the influence of molecular weight on the viscosity of polyethylene glycol (PEG). It was established that as the molecular weight of PEG decreased, there was a reduction in viscosity, which mirrors the results obtained in this study [53].

#### 3.3.2. Differential Scanning Calorimetry

To evaluate the thermal properties, DSC analysis was carried out on the polymer, two inorganic fillers, and various composites. Due to the semi-crystalline nature of PEBA, it is of interest to understand how the addition of inorganic fillers affects the crystallinity and other thermal properties. To provide a comprehensive overview of the thermal characteristics, PEBA100, the various composites of BaSO_4_ and BiOCl, and the inorganic fillers were all analyzed via DSC. Upon initial review of the graphs, it could be seen that while there is minimal movement in the temperature at which endo and exothermic peaks occur in PEBA/BaSO_4_ composites, there were greater shifts in the temperature at which the exothermic peaks occurred for the PEBA/BiOCl composites. These shifts in temperature for exothermic peaks are caused by crystallization of the polymer, and these occur during cooling, with crystallization occurring at a greater temperature. This signifies a reduction in the degree of crystallinity of the polymer., suggesting that the PEBA/BiOCl composites have a lower degree of crystallinity than both the polymer and PEBA/BaSO_4_ composites. Twin crystallization peaks were observed during cooling of the PEBA/BiOCl composites; however, these were not present in the thermograms of PEBA/BaSO_4_. One aspect to consider when understanding the effects of filler on the degree of crystallinity is the nucleating that occurs in the polymer due to the incorporation of fillers. Pingping and Dezhu conducted similar trials on the effects of various addition rates of calcium carbonate (CaCO_3_) on the cold crystallization peaks of poly (ethylene terephthalate) (PET). It was found that with an increase in the addition of CaCO_3_, there was a clear reduction in the degree of crystallinity of the composite. It was summarized that the amorphous regions between spherulites decrease considerably with the addition of CaCO_3_. These results conform well with the mechanism of the effect of nucleation of CaCO_3_ on the PET crystallization process [54]. It could be said that the BaSO_4_ filler induces a greater degree of crystallization in comparison to the BiOCl, due to an enhanced level of nucleation.

Table 8 provides the values obtained from the DSC analysis. It can be seen in the table that while there was minimal change in melt temperature of the composites in comparison to the polymer (PEBA100), there was in fact a change in the melting enthalpies and the crystallization temperature, where subsequently, the degree of crystallinity has shifted in the composites. The reductions in degree of crystallinity for the PEBA/BaSO_4_-10, -20, and -30 composites were 10.1%, 30%, and 28.9% respectively. On the other hand, the PEBA/BiOCl-10, -20, and -30 composites experienced a 9.96%, 40.15%, and 37.43% reduction in crystallinity respectively when compared to the PEBA100, showing that the PEBA/BiOCl composites experienced a greater reduction in crystallinity in comparison to the comparatively loaded PEBA/BaSO_4_ composites. These results may signify a greater degree of crystallinity of the BaSO_4_ inorganic filler itself when compared to the BiOCl. This was also investigated by performing DSC on the fillers; however, there were no observed endothermic or exothermic peaks, possibly due to the relatively low temperatures at which the testing was conducted. It may be a case whereby the BaSO_4_ has overlapping crystallinity when incorporated into the polymer, therefore obtaining a more crystalline composite.

Liu et al. assessed the effects of melt compounding and the addition of various loading of inorganic fillers on the thermal properties of PLA. It was established that first and foremost, the shear introduced into the polymer during melt compounding had caused the degree of crystallinity to decrease when compared to the PLA. The observed reduction in the degree of crystallinity was significant, at 64.3% [55]. This reduction in crystallinity showed the effects of shear imparted into the polymer, which, when compared to the results obtained in this study, were mirrored by the reduction in the degree of crystallinity of the melt compounding composites, however, not to such an extent. Similarly, in this study, it was found that as the density of the inorganic fillers increased, the degree of crystallinity decreased significantly. While the trend was similar, it was not to such an extent as observed by Liu et al. In this study, the observed difference between the two inorganic fillers at the respective loadings was 0.61%, 12.63%, and 12.33%. It was also noted that while the degree of crystallinity decreased with the increase in filler content, they increased slightly at the highest level of loading when compared to the respective middle loading (20 wt.%).

Khalaf conducted DSC analysis on composites with various loading levels of filler. It was established that as the percentage of filler loading increased, the degree of crystallinity decreased. This was attributed to the lower free volume within the polymer due to the greater volume of filler, thereby reducing the crystallization ability of the polymer [56]. Changes in a polymer’s morphology such as degree of crystallinity can lead to phenomena such as shrinkage in parts. When designing a medical device product, it is important to understand the crystallinity of the polymer or polymer composite, so that accommodations may be made for shrinkage that may occur during process, either post-extrusion, or while molding.

### 3.4. Physical Properties

#### 3.4.1. Density

Density is a crucial characteristic to consider when evaluating composites of various filler content for medical devices. When comparing various filler types of dissimilar densities, it is important to look at the densities of the final composites, showing clearly the effects of loading rates a particular filler into a polymer system. In this case, where BaSO_4_ and BiOCl were incorporated into the polymer at various rates, with such as difference in density of the inorganic fillers, the densities of each formulation of composites were analyzed. Knowing the density of a material or composite is important as it provides critical understanding about how the material would behave during processing, but also for design of a medical device. It has also been shown that incorporation of fillers of varying densities at similar weight percentages may have differing effects on the reinforcement of the base polymer. As such, modification of the weight percentage of filler added may require alteration [57]. The reinforcement effect herein was, however, not affected by the differing densities of the filler.

Figure 9 displays the densities of the various composites produced during compounding while Table 9 shows a comparison of theoretic and actual densities. All experimentally obtained values were within 0.01% of the theoretical values, showing the effectiveness of the extrusion and molding processes in creating uniform blends. It is clear from the graph that with the incorporation of both inorganic fillers into the respective composites, the densities increased from that of the polymer. When comparing the composites against PEBA100 (1.01 g/cm^3^), there was an increase of 8.73%, 14.65%, and 23.23% for the PEBA/BaSO_4_-10 (1.107 g/cm^3^), -20 (1.183 g/cm^3^), and -30 (1.316 g/cm^3^) composites, respectively. Similarly, a 9.39%, 16.67%, and 26.85% increase was observed for PEBA/BiOCl-10 (1.115 g/cm^3^), -20 (1.212 g/cm^3^), and -30 (1.381 g/cm^3^), respectively. It was established that the PEBA/BiOCl composites were denser than the PEBA/BaSO_4_, with an increase of 0.72%, 2.37%, and 4.71% at the respective increases in filler loading. It was expected that the BiOCl composites would have a greater density at the respective loadings than the PEBA/BaSO_4_ composites.

#### 3.4.2. Ash Content

Ash content is an excellent method for evaluation of the content of inorganic filler in a polymer composite after compounding. While this method does not offer as much in-depth data, it does provide an indication of the level of filler within the composite. Figure 10 displays the results obtained from the ash content test. It is clear from the graph that the desired filler loadings for the composites were achieved by compounding. There was some variation in the addition of filler for PEBA/BaSO_4_-10, -20, and -30 of 0.5%, 0.84%, and 0.07%, respectively. Similarly, the discrepancy in filler loading of PEBA/BiOCl-10, -20, and -30 was 1.66%, 0.15%, and 0.76%, respectively, displaying the accuracy of feeding during the compounding process.

#### 3.4.3. Radiopacity Contrast

For the development of effective medical tubing, it is imperative to find a balance between the mechanical durability and strength of the tubing with the contrast observed under X-ray. Figure 11 displays the prepared composite samples observed under X-ray fluoroscopy. The limited visibility of PEBA100 displays the requirement for the incorporation of radiopaque fillers, as the sample is essentially invisible under X-ray examination. With the incremental addition of filler, the radiocontrast begins to increase. This increase is significantly more pronounced for the samples incorporating BiOCl compared to those using BaSO_4_. BaSO_4_ continues to be the most widely used agent for enhancing the radiopaqueness of polymers [30,58,59]. The high loadings required to obtain this contrast, up to 60 wt.%, may have drawbacks on the ductility of medical tubing and have cost implications. Thus, bismuth-based compounds have become increasingly competitive, especially BiOCl [60]. Bismuth, as the heaviest non-radioactive metal, displays excellent X-ray imaging capabilities, based on having an increased attenuation coefficient in comparison to barium and iodine [61,62].

As can be observed in Figure 11, PEBA70BiOCl30 appears roughly twice as dark as its BaSO_4_ counterpart. The increased density of the BiOCl in comparison to the BaSO_4_ has been shown by additional authors as presenting a doubled level of radiopacity [63,64]. The significant enhancement in radiopacity at lower loading levels, while maintaining the mechanical properties of the PEBA, make BiOCl an excellent candidate for the manufacture of medical tubing. The lower loading levels allow for a smooth surface finish, resulting in easier insertion in the patient and reducing the overall risk of thrombus formation [30].

## 4. Conclusions

This study carried out an investigation into the effect of adding inorganic fillers, specifically barium sulphate and bismuth oxychloride, on the processing and material properties of PEBA while maintaining a high degree of X-ray contrast. PEBAX, due to its excellent mechanical and thermal properties, is commonly used in medical tubing applications. Both BaSO_4_ and BiOCl are favored in medical applications due to their excellent X-ray contrast capabilities. The research analyzed the impact of filler addition through melt processing followed by injection molding of ASTM standard test specimens followed by analysis of thermal, mechanical, and physical properties as well as observing the X-ray contrast of samples via fluoroscopy.

The filler type and loading concentration influenced the material processing parameters with BiOCl composites displaying higher torque values leading to a reduction in screw speeds in order to maintain a stable process compared to the BaSO_4_ composites. The thermal properties were additionally impacted with BiOCl, exhibiting higher melt temperatures owing to the greater thermal conductivity exerted by the BiOCl. The physical appearances of the composites varied, with BiOCl composites appearing more opaque and whiter, whereas the BaSO_4_ displayed a more cream-colored tint.

During the injection molding of the composite samples, the increased filler content and associated changes in viscosity and density led to changes in both plasticizing time and melt cushion. The reduction in viscosity shown by the higher filler levels lends credence to the processing causing a slight molecular weight decrease caused by processing shear. With regards to tensile properties, the BaSO_4_ composites displayed a more pronounced reduction in tensile strength though they did display increased stiffness. The BaSO_4_ samples showed a higher stiffness than BiOCl, though BiOCl composites displayed greater flexural moduli, indicative of greater stiffness without causing brittleness. This would lend the BiOCl composites to being good candidates for medical tubing where both a high degree of stiffness and flexibility are required.

In conclusion, this research shows that BiOCl is a better candidate for the manufacture of radiopaque tubing, as it shows a more balanced enhancement of properties without negatively affecting mechanical performance, making it preferable for medical tubing requiring radiopacity, flexibility, and strength. This underscores the importance of optimizing filler loading concentration and processing conditions for medical tubing production.

## Figures and Tables

**Figure 1 polymers-16-03220-f001:**
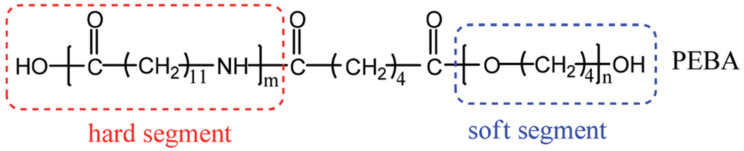
PEBA chemical structure.

**Figure 2 polymers-16-03220-f002:**
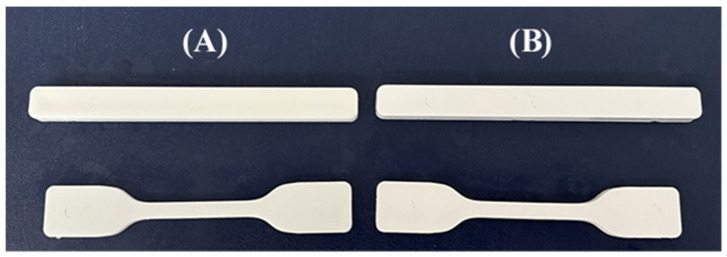
Tensile and impact test specimen, (**A**) 30 wt.% BaSO_4_ and (**B**) 30 wt.% BiOCl.

**Figure 3 polymers-16-03220-f003:**
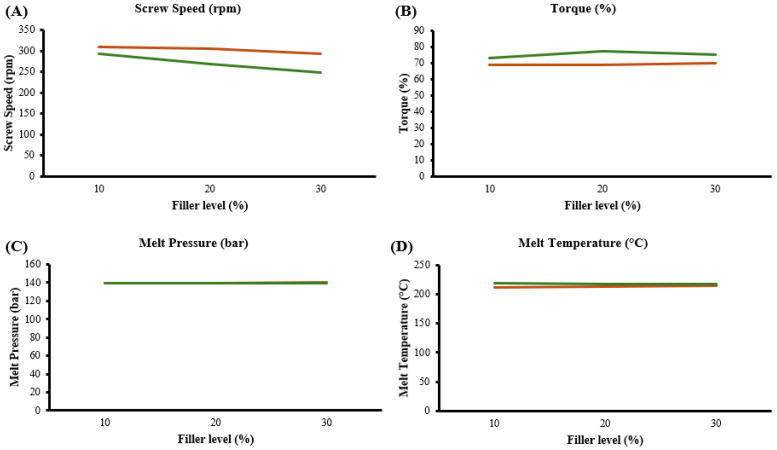
Twin-screw compounding observations, (**A**) screw speed, (**B**) torque, (**C**) melt pressure, and (**D**) melt temperature; BaSO_4_-filled Pebax^®^ 6333 SA01 MED (orange), BiOCl-filled Pebax^®^ 6333 SA01 MED (green).

**Figure 4 polymers-16-03220-f004:**
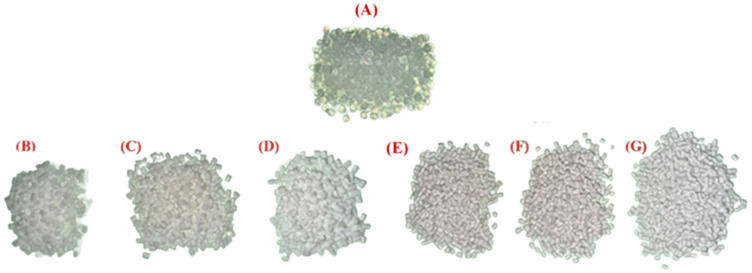
Pellets pre- and post-compounding, (**A**) PEBA-100, (**B**) PEBA/BaSO_4_-10, (**C**) PEBA/BaSO_4_-20, (**D**) PEBA/BaSO_4_-30, (**E**) PEBA/BiOCl-10, (**F**) PEBA/BiOCl-20, (**G**) PEBA/BiOCl-30.

**Figure 5 polymers-16-03220-f005:**
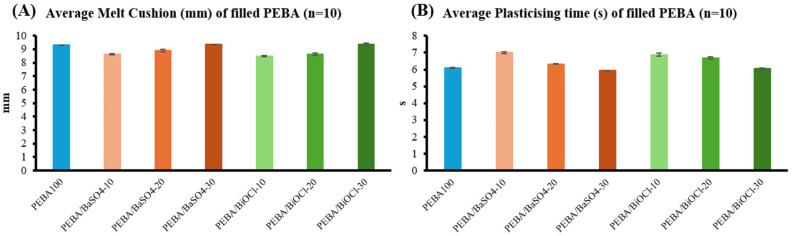
Injection-molding observations, (**A**) average melt cushion, (**B**) average plasticizing time; Pebax^®^ 6333 SA01 MED (blue), BaSO_4_-filled Pebax^®^ 6333 SA01 MED (orange), BiOCl-filled Pebax^®^ 6333 SA01 MED (green).

**Figure 6 polymers-16-03220-f006:**
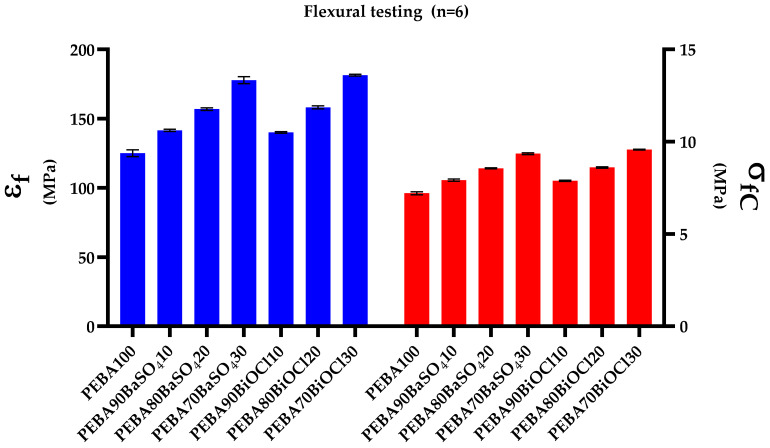
Flexural properties of all tested samples (n = 6). ε_f_: flexural modulus (blue) and σ_fc_: flexural strength (red).

**Figure 7 polymers-16-03220-f007:**
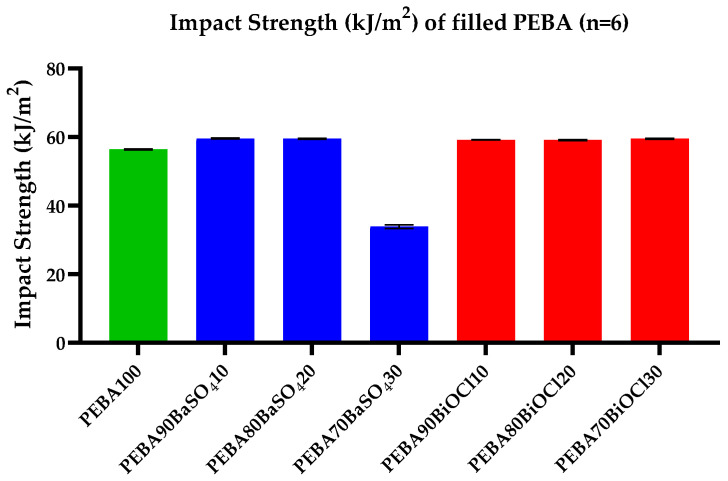
Impact properties of all tested samples (n = 6): Pebax^®^ 6333 SA01 MED (blue), BaSO_4_-filled Pebax^®^ 6333 SA01 MED (orange), BiOCl-filled Pebax^®^ 6333 SA01 MED (green).

**Figure 8 polymers-16-03220-f008:**
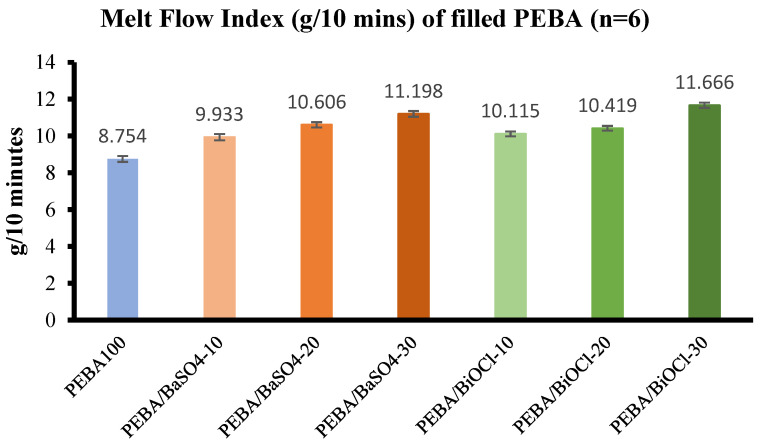
Melt flow analysis: Pebax^®^ 6333 SA01 MED (blue), BaSO_4_-filled Pebax^®^ 6333 SA01 MED (orange), BiOCl-filled Pebax^®^ 6333 SA01 MED (green).

**Figure 9 polymers-16-03220-f009:**
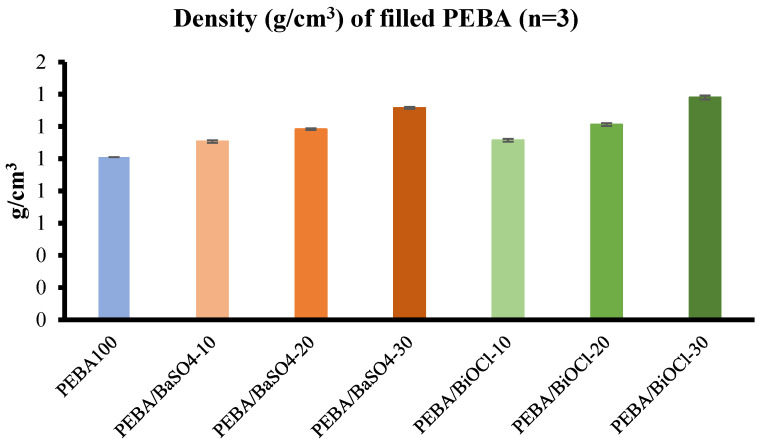
Density analysis: Pebax^®^ 6333 SA01 MED (blue), BaSO_4_-filled Pebax^®^ 6333 SA01 MED (orange), BiOCl-filled Pebax^®^ 6333 SA01 MED (green).

**Figure 10 polymers-16-03220-f010:**
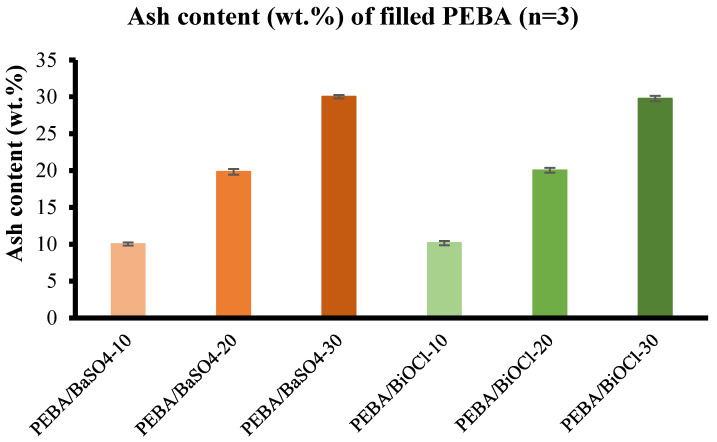
Ash content analysis: BaSO_4_-filled Pebax^®^ 6333 SA01 MED (orange), BiOCl-filled Pebax^®^ 6333 SA01 MED (green).

**Figure 11 polymers-16-03220-f011:**
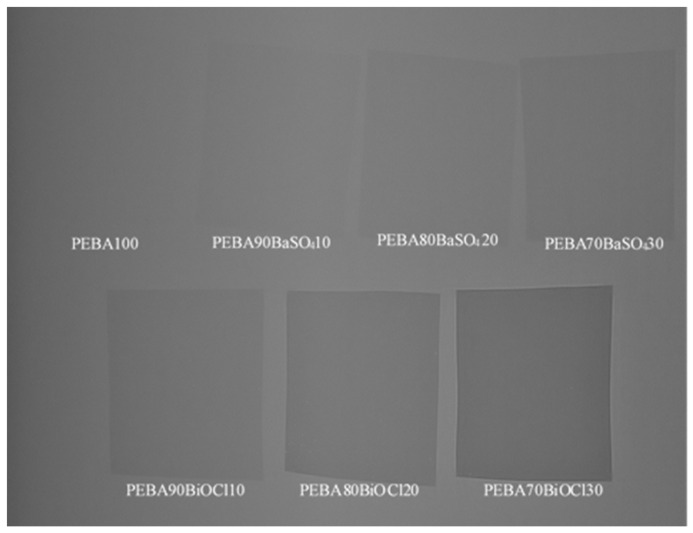
Images of the prepared composites under X-ray fluoroscopy.

**Table 1 polymers-16-03220-t001:** Various Pebax^®^ grades properties [16,17,18,19].

Properties	Units	Pebax^®^ 2533 SA01 MED	Pebax^®^ 4033 SA01 MED	Pebax^®^ 6333 SA01 MED	Pebax^®^ 7233 SA01 MED
Shore A hardness	-	74	89	~100 (estimate)	>100 (estimate)
Shore D hardness	-	~25 (estimate)	35	58	61
Tensile modulus	MPa	12	73	307	510
Charpy notched impact strength, +23 °C	kJ/m^2^	No break	No break	No break	15

**Table 3 polymers-16-03220-t003:** Pebax^®^ 6333 SA01 MED material properties [18]. Cond (conditioned) for at least 16 h at 23 ± 2 °C and 50 ± 10%RH (relative humidity).

Property	Dry/Cond	Unit	Test Standard
Tensile modulus	307/240	MPa	[33]
Yield stress	19/18	MPa	
Yield strain	22/22	%	
Nominal strain at break	50/>50	%	
Shore D hardness, after 15 s	58	-	[34]
Charpy notched impact strength, +23 °C	-/No Break	kJ/m^2^	[35]
Density	1.01/-	g/cm^3^	[36]

**Table 4 polymers-16-03220-t004:** Composition of materials produced for this study. Wt.%—weight percentage.

Sample Name	PEBA (wt.%)	BaSO_4_ (wt.%)	BiOCl (wt.%)
PEBA100	100	0	0
PEBA/BaSO_4_10	90	10	0
PEBA/BaSO_4_20	80	20	0
PEBA/BaSO_4_30	70	30	0
PEBA/BiOCl10	90	0	10
PEBA/BiOCl20	80	0	20
PEBA/BiOCl30	70	0	30

**Table 5 polymers-16-03220-t005:** Extrusion temperatures utilized for the preparation of PEBAX/filler composites.

Heating Zone	Parameter (°C)
Zone 1	80
Zone 2	160
Zone 3	180
Zone 4	180
Zone 5	180
Zone 6	180
Zone 7	180
Zone 8	190
Zone 9	190
Zone 10	190

**Table 6 polymers-16-03220-t006:** Tensile properties of all tested samples (n = 6). Σ: tensile strength, σ: stress at break, ε: elongation at break, and E: tensile modulus.

Sample	Σ (MPa)	ε (%)	E (MPa)
PEBA100	39.78 ± 0.69	330.35 ± 5.06	126.96 ± 28.57
PEBA90BiOCl10	39.67 ± 0.89	310.78 ± 8.50	159.90 ± 25.44
PEBA80BiOCl20	40.67 ± 0.79	311.90 ± 7.05	153.92 ± 33.33
PEBA70BiOCl30	40.66 ± 0.16	314.79 ± 3.84	189.92 ± 15.94
PEBA90BaS0_4_10	36.95 ± 1.57	303.19 ± 16.06	134.79 ± 24.17
PEBA80BaS0_4_20	36.78 ± 0.69	307.83 ± 6.23	155.26 ± 36.71
PEBA70BaS0_4_30	32.88 ± 1.47	281.40 ± 16.72	172.46 ± 58.18

**Table 7 polymers-16-03220-t007:** Flexural properties of the tested PEBA-based composites. ε_f_: flexural modulus and σ_fc_: flexural strength.

	ε_f_ (MPa)	% Increase	σ_fc_ (MPa)	% Increase
PEBA100	125.01 ± 2.44	-	7.20 ± 0.08	-
PEBA90BaSO_4_10	141.50 ± 0.78	13.19	7.92 ± 0.06	10.00
PEBA80BaSO_4_20	156.82 ± 0.91	25.45	8.55 ± 0.03	35.00
PEBA70BaSO_4_30	177.72 ± 2.52	42.16	9.35 ± 0.05	29.86
PEBA90BiOCl10	140.01 ± 0.57	12.00	7.88 ± 0.03	9.44
PEBA80BiOCl20	158.08 ± 1.05	26.45	8.60 ± 0.04	19.44
PEBA70BiOCl30	181.35 ± 0.65	45.07	9.57 ± 0.02	32.92

**Table 8 polymers-16-03220-t008:** Thermal transitions identified from resultant thermograms. Χ_c_ = (ΔH/ΔH*); ΔH is the melting enthalpy calculated through integration of the melting peak determined via DSC; ΔH* = 65 J.g^−1^ is the melting enthalpy of a 100% crystalline Pebax^®^.

Sample	T_m_ (°C)	ΔH_m_ (J g^−1^)	T_c_ (°C)	ΔH_cc_ (J g^−1^)	Χ_c_ (%)
PEBA100	172.30	58.12	133.67	62.29	89.41
PEBA/BaSO_4_10	172.59	52.25	136.17	54.95	80.38
PEBA/BaSO_4_20	172.65	40.70	136.90	44.69	62.62
PEBA/BaSO_4_30	173.71	41.35	136.89	45.89	63.61
PEBA/BIOCl10	173.53	52.33	144.03	52.54	79.89
PEBA/BIOCl20	173.36	34.79	144.01	38.22	54.71
PEBA/BIOCl30	171.44	36.39	143.34	39.06	55.77

**Table 9 polymers-16-03220-t009:** Comparison of theoretical densities of the composites prepared and the values obtained experimentally. ρtheor: theoretical density and ρact: actual density.

	ρ_theor_ (g/cm^3^)	ρ_act_ (g/cm^3^)	Deviation (%)
PEBA100	1.010	1.010	0.0100
PEBA/90BaSO_4_10	1.095	1.107	0.0101
PEBA/80BaSO_4_20	1.195	1.183	0.0099
PEBA/70BaSO_4_30	1.316	1.316	0.0100
PEBA/90BiOCl10	1.106	1.115	0.0101
PEBA/80BiOCl20	1.222	1.212	0.0099
PEBA/70BiOCl30	1.366	1.381	0.0101

## Data Availability

The original contributions presented in the study are included in the article, further inquiries can be directed to the corresponding author.

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
