# Peer review of "Co-Optimization of Mechanical Properties and Radiopacity Through Radiopaque Filler Incorporation for Medical Tubing Applications"

_polymers, 2024, doi:10.3390/polym16223220_

Round 1

Reviewer 1 Report

Comments and Suggestions for Authors

1. Abstract and conclusions are written very exhaustively  , need to be shorten

2. Authors recommended to conduct DSC  for second heating and cooling curves in order  understand the effect of plasticizers in figure 7

Author Response

Comment 1: Abstract and conclusions are written very exhaustively, need to be shorten

Response to Comment 1: Thank you for the comment. We have revised and shortened both abstract and conclusion.

Comment 2: Authors recommended to conduct DSC for second heating and cooling curves in order understand the effect of plasticizers in figure 7

Response to Comment 2: Thank you for your comment. While yes that would be a good idea it is outside of the purpose of this testing where we are looking at the effect of the filler loading and variety on the thermal properties.

Reviewer 2 Report

Comments and Suggestions for Authors

This paper is devoted to the joint optimization of mechanical properties and radiopacity for medical tubing by incorporating a radiopaque filler. The effect of inorganic fillers such as Pebax® 6333 SA01 MED (PEBA100), barium sulfate (PEBA/BaSO4) and bismuth oxychloride (PEBA/BiOCl) with different loads. The technological, mechanical, thermal and physical properties of the polymers were investigated. Optimizations of the filler content and processing conditions were performed to achieve the desired properties of the composite for common medical applications was demonstrated. The work is relevant and can be accepted for publication, however, there are few issues:

1. Line 659: Melt flow Indexes [g/10 min] (Figure 9) and densities [g/cm3] (Figure 12) have similar trends depending on the concentration and type of filler. Will the volumetric melt flow index remain constant?

2. Line 769: Are there any typos in the ΔHcc values for PEBA/BIOCl10? The crystallization peak areas in Figure 11 does not look that different for PEBA/BiOCl-10, PEBA/BiOCl-20, and PEBA/BiOCl-30.

3. Lines 107, 121, 786. There is a phrase "Error! Reference source not found." Please check?

4. A good rule of thumb for papers is that more than 50% of the cited literature should be published in the last 5 years. In this paper, the list of cited literature from the last 5 years is less than 50%. The reader may wonder about the relevance of the topic of this paper and the quality of the choice of literature.

Author Response

Comment 1: Line 659: Melt flow Indexes [g/10 min] (Figure 9) and densities [g/cm3] (Figure 12) have similar trends depending on the concentration and type of filler. Will the volumetric melt flow index remain constant?

Response to Comment 1: Even if the densities and MFIs show similar trends the volumetric melt ow index may not remain constant. The MFI in terms of volume is influenced by both the mass flow rate and the material density. If the density of the material changes with different filler concentrations or types, the volumetric MFI will likely vary accordingly. For example, if increasing the filler concentration results in a higher density, the same mass-based MFI (g/10 min) would translate to a lower volumetric MFI, as the material would occupy less volume per unit mass. Conversely, if density decreases with a particular filler type or concentration, the volumetric MFI would increase.

Comment 2: Line 769: Are there any typos in the ΔHcc values for PEBA/BIOCl10? The crystallization peak areas in Figure 11 does not look that different for PEBA/BiOCl-10, PEBA/BiOCl-20, and PEBA/BiOCl-30.

Response to Comment 2: We have reviewed and corrected the values obtained for the DSCs

Comment 3: Lines 107, 121, 786. There is a phrase "Error! Reference source not found." Please check?

Response to Comment 3: Thank you for pointing this out. These errors have been rectified

Comment 4: A good rule of thumb for papers is that more than 50% of the cited literature should be published in the last 5 years. In this paper, the list of cited literature from the last 5 years is less than 50%. The reader may wonder about the relevance of the topic of this paper and the quality of the choice of literature.

Response to Comment 4: Thank you for this comment. We agree that the cited sources were outside of the 5 year period but still relevant sources of information from which to cite. Regardless we have bolstered the paper with further references within the past 5 years.

Reviewer 3 Report

Comments and Suggestions for Authors

This work can be interesting to readers who are interested in radiopaque materials in biomedical applications. However, the current version is not acceptable for publication. Please consider addressing the following comments.

Major issues:

1. In Figure 6, I recommend the authors replace Figure 6 with a table where the readers can see the differences while comparing the results. 

Also, note that any conclusion drawn from the Figures should be based on statistically meaningful data. For example, the authors compared both strengths of lines 454 to 456, it is challenging for readers to see the major difference from Figure 6A while comparing the strengths of PEBA/BaSO4-10 and  PEBA/BiOCl-10 to that of PEBA100. It seems that the strengths of PEBA/BiOCl-10 and PEBA100 do not have statistical differences (both errors overlap). It also seems to me the error bar of PEBA/BaSO4-10 overlaps with that of PEBA100, so there seems no statistical difference. 

Similar issues can be found in the discussion of Figures 6B, 6C, and 6D.

Please refine the results in Figure 6 and discuss the major findings. Please also make a concise discussion of the results of Figure 7.

2. The conclusion part is filled with too much redundant information. I recommend refining the major findings and re-write the conclusion in 2-3 paragraphs. 

3. The authors claimed the materials are radiopaque. They should perform the X-ray image analysis of models or samples and compare the results to the existing commercial samples or biological samples (such as bone). 

The authors tested different filler concentrations.  For the x-ray contrast, to what degree the filler concentration should be good enough? 

Minor issues: 

1. Line 302, equation 6, I think density =mass/volume. Also, in line 308, please use rho instead of P for density.

2. Figure 16A and 16B in lines 397 and 398 should be Figures 5A and 5B.

3. Similarly, in line 628, Figure 19 should be Figure 8. 

4. please correct the reference error messages in lines 107, 121, 786.

5. Please leave Table 2 and Table 4 on the same page, respectively.

6. Line 271, add a blank spacer between 27mm. Similar in line 206.

7. The text in Pages 20 and 21 should be justify aligned.

Author Response

Comment 1: In Figure 6, I recommend the authors replace Figure 6 with a table where the readers can see the differences while comparing the results. 

Response to Comment 1: Thank you for your feedback, Figure 6 has been replaced with Table 6.

Also, note that any conclusion drawn from the Figures should be based on statistically meaningful data. For example, the authors compared both strengths of lines 454 to 456, it is challenging for readers to see the major difference from Figure 6A while comparing the strengths of PEBA/BaSO4-10 and  PEBA/BiOCl-10 to that of PEBA100. It seems that the strengths of PEBA/BiOCl-10 and PEBA100 do not have statistical differences (both errors overlap). It also seems to me the error bar of PEBA/BaSO4-10 overlaps with that of PEBA100, so there seems no statistical difference. 

Similar issues can be found in the discussion of Figures 6B, 6C, and 6D.

Please refine the results in Figure 6 and discuss the major findings. Please also make a concise discussion of the results of Figure 7.

Comment 2: The conclusion part is filled with too much redundant information. I recommend refining the major findings and re-write the conclusion in 2-3 paragraphs. 

Response to Comment 2: Thank you for your comment, the conclusion has been summarised into 4 paragraphs.

Comment 3: The authors claimed the materials are radiopaque. They should perform the X-ray image analysis of models or samples and compare the results to the existing commercial samples or biological samples (such as bone). 

The authors tested different filler concentrations.  For the x-ray contrast, to what degree the filler concentration should be good enough? 

Response to Comment 3: Thank you for your comment. We have taken this on board with the inclusion of X-ray medical imaging similar to fluoroscopy (2.6.3)

Minor issues: 

Minor issue 1: Line 302, equation 6, I think density =mass/volume. Also, in line 308, please use rho instead of P for density.

Response: This has been rectified

Minor issue 2: Figure 16A and 16B in lines 397 and 398 should be Figures 5A and 5B.

Response: This has been rectified

Minor issue 3: Similarly, in line 628, Figure 19 should be Figure 8. 

Response: This has been rectified

Minor issue 4: please correct the reference error messages in lines 107, 121, 786.

Response: This has been rectified

Minor issue 5: Please leave Table 2 and Table 4 on the same page, respectively.

Response: This has been rectified

Minor issue 6: Line 271, add a blank spacer between 27mm. Similar in line 206.

Response: This has been rectified

Minor issue 7: The text in Pages 20 and 21 should be justify aligned.

Response: This has been rectified

Round 2

Reviewer 3 Report

Comments and Suggestions for Authors

The authors addressed well to the reveiwers' comments. 

A few minor changes are required.

1. Please remove pages 5, 7 (blank pages)

2. Fig 6, please adjust the y-axis titles. stress and strain symbols are not clear. 

Author Response

Comment 1: Please remove pages 5, 7 (blank pages)

Response to Comment 1: Thank you for pointing this out, these blank pages have been removed

Comment 2: Fig 6, please adjust the y-axis titles. stress and strain symbols are not clear

Response to Comment 2: Thank you for the feedback, we have adjusted the font size of the axis label to make it easier to read